# Expression Profile and Prognostic Value of Wnt Signaling Pathway Molecules in Colorectal Cancer

**DOI:** 10.3390/biomedicines9101331

**Published:** 2021-09-27

**Authors:** Yung-Fu Wu, Chih-Yang Wang, Wan-Chun Tang, Yu-Cheng Lee, Hoang Dang Khoa Ta, Li-Chia Lin, Syu-Ruei Pan, Yi-Chun Ni, Gangga Anuraga, Kuen-Haur Lee

**Affiliations:** 1National Defense Medical Center, Department of Medical Research, School of Medicine, Tri-Service General Hospital, Taipei 11490, Taiwan; qrince@yahoo.com.tw; 2PhD Program for Cancer Molecular Biology and Drug Discovery, College of Medical Science and Technology, Taipei Medical University, Taipei 11031, Taiwan; chihyang@tmu.edu.tw (C.-Y.W.); yeas0310@hotmail.com (W.-C.T.); 3Graduate Institute of Cancer Biology and Drug Discovery, College of Medical Science and Technology, Taipei Medical University, Taipei 11031, Taiwan; m654108001@tmu.edu.tw (L.-C.L.); panray8802069487@gmail.com (S.-R.P.); ckni1012@gmail.com (Y.-C.N.); 4Graduate Institute of Medical Sciences, College of Medicine, Taipei Medical University, Taipei 11031, Taiwan; yclee0212@tmu.edu.tw; 5PhD Program for Cancer Molecular Biology and Drug Discovery, College of Medical Science and Technology, Taipei Medical University and Academia Sinica, Taipei 11031, Taiwan; d621109004@tmu.edu.tw (H.D.K.T.); g.anuraga@unipasby.ac.id (G.A.); 6Department of Statistics, Faculty of Science and Technology, Universitas PGRI AdiBuana, Surabaya 60234, East Java, Indonesia; 7Cancer Center, Wan Fang Hospital, Taipei Medical University, Taipei 11031, Taiwan

**Keywords:** colorectal cancer, biomarker, prognosis, CTNNB1, Wnt/beta-catenin

## Abstract

Colorectal cancer (CRC) is a heterogeneous disease with changes in the genetic and epigenetic levels of various genes. The molecular assessment of CRC is gaining increasing attention, and furthermore, there is an increase in biomarker use for disease prognostication. Therefore, the identification of different gene biomarkers through messenger RNA (mRNA) abundance levels may be useful for capturing the complex effects of CRC. In this study, we demonstrate that the high mRNA levels of 10 upregulated genes (*DPEP1*, *KRT80*, *FABP6*, *NKD2*, *FOXQ1*, *CEMIP*, *ETV4*, *TESC*, *FUT1*, and *GAS2*) are observed in CRC cell lines and public CRC datasets. Moreover, we find that a high mRNA expression of DPEP1, NKD2, CEMIP, ETV4, TESC, or FUT1 is significantly correlated with a worse prognosis in CRC patients. Further investigation reveals that CTNNB1 is the key factor in the interaction of the canonical Wnt signaling pathway with 10 upregulated CRC-associated genes. In particular, we identify *NKD2*, *FOXQ1*, and *CEMIP* as three CTNNB1-regulated genes. Moreover, individual inhibition of the expression of three CTNNB1-regulated genes can cause the growth inhibition of CRC cells. This study reveals efficient biomarkers for the prognosis of CRC and provides a new molecular interaction network for CRC.

## 1. Introduction

Colorectal cancer (CRC) is one of the most malignant cancers and the third most prevalent cancer worldwide [1]. By 2030, the global CRC burden is expected to increase by 60%, with >2.2 million new cases and 1.1 million deaths [2]. Detection of CRC is mainly still based on the fecal occult blood test (FOBT) and colonoscopy. Development and progression of CRC involves changes in the genetic and epigenetic levels of different genes [3]. Although several biomarkers (adenomatous polyposis coli (APC), CTNNB1, and p53) are used to detect CRC, these biomarkers are not sufficiently sensitive and specific [4]. As molecules involved in CRC progression are not completely understood, further research is required to discover and investigate effective biomarkers for the diagnosis and prognosis of CRC [5].

Gene expression profiles have been broadly used to identify novel biomarkers through high-throughput technology [6]. However, CRC is a heterogeneous disease, with different subtypes identified and characterized by specific molecular alterations [7]. Recently, in the context of CRC, molecular assessments have received growing attention, with a parallel increase in biomarker use for prognostication [8]. Therefore, identifying different gene biomarkers may be beneficial for capturing the complex effects of heterogeneous diseases, such as CRC, on messenger RNA (mRNA) abundance [9]. Our recent study [10] showed that the metallothionein genes play a role in the CRC prognosis which bellowed to a metallothionein gene family, and this family contains the top 10 downregulated genes in CRC tissues according to a Gene Expression Omnibus (GEO) dataset (GSE21815) analysis [11]. In addition, our recent study identified the top 10 upregulated genes in CRC tissues compared with normal colorectal tissues (GSE21815) (our unpublished data from [10]). Although some of these top 10 upregulated CRC-associated genes are involved in CRC progression [12], the expression levels, prognostic relevance, and molecular mechanism of these genes in relation to CRC remain unclear.

Thus, in the present study, we demonstrate that the mRNA levels of 10 upregulated CRC-associated genes (*DPEP1*, *KRT80*, *FABP6*, *NKD2*, *FOXQ1*, *CEMIP*, *ETV4*, *TESC*, *FUT1*, and *GAS2*) exhibit simultaneous upregulation in CRC cell lines and public CRC datasets. In addition, we find that the mRNA expression levels of 10 upregulated CRC-associated genes are significantly and positively related to the CRC stage. We further investigate the prognostic significance of these genes in CRC outcomes. Further investigations reveal that the key gene involved in the canonical Wnt signaling pathway that interacts with the 10 upregulated CRC-associated genes is CTNNB1. We observe *NKD2*, *FOXQ1*, and *CEMIP* to be CTNNB1-regulated genes, and individual inhibition of the expression of three CTNNB1-regulated genes can cause growth inhibition of CRC cells.

## 2. Materials and Methods

### 2.1. Cell Lines, Culture Method, and Reagents

Human CRC cell line HCT116 was provided by Professor YW Cheng (Graduate Institute of Cancer Biology and Drug Discovery, Taipei Medical University). The cell line was cultured in RPMI-1640, supplemented with 10% fetal bovine serum, and 1% penicillin–streptomycin (50 U/mL, 50 μg/mL, respectively) (all from Thermo Fisher Scientific, Waltham, MA, USA), and were maintained at 37 °C in a humidified atmosphere containing 5% CO_2_.

### 2.2. Cell Transfections

CTNNB1, NKD2, FOXQ1, CEMIP, and scrambled negative control small interfering (si)RNAs were purchased from Invitrogen (Carlsbad, CA, USA), and transfected into cells using the jetPRIME transfection reagent (Polyplus-transfection, New York, NY, USA) according to the manufacturer’s instructions. Sequences of the siRNAs are described in the Appendix A.

### 2.3. Cell Viability Assay

Cell viability was determined with the crystal violet-staining method. In brief, the oligonucleotide (50 or 100 nM) was introduced into 5 × 10^5^ dissociated cells by the jetPRIME transfection reagent according to the manufacturer’s instructions. Next, cells were plated in 96-well plates at 3000 cells/mL after transfection with control siRNA or siRNA of *NKD2*, *FOXQ1*, and *CEMIP* for 48 h. After cells had grown for 48 h, cells were stained with 0.5% crystal violet for 10 min at room temperature. Next, the plates were washed with tap water three times. After drying, cells were lysed with a 0.1 M sodium citrate solution (Sigma–Aldrich, St. Louis, MO, USA), and the absorbance was measured at 550 nm on a microplate reader.

### 2.4. Focal Formation Assays

HCT116 cells were seeded at 1000 cells/well in six-well dishes and grown overnight after transfection with control siRNA or siRNA of *NKD2*, *FOXQ1*, and *CEMIP* for 48 h, respectively. The medium was changed every 3 days. After 7 days, the cells were fixed and stained with 0.5% crystal violet. Foci of >5 mm in size were counted, and average focal counts and standard deviations (SDs) were calculated.

### 2.5. Quantitative Reverse-Transcription Polymerase Chain Reaction

Total RNA was extracted from CRC cell line by using a Qiagen RNeasy kit (Qiagen, Valencia, CA, USA) and QIAshredder columns according to the manufacturer’s instructions (Qiagen, Valencia, CA, USA). One microgram of total RNA was reverse-transcribed to complementary DNA (cDNA) by using a reaction-ready First Strand cDNA Synthesis Kit (SABiosciences, Frederick, MD, USA). To detect the mRNA expression levels of NKD2, FOXQ1, and CEMIP genes, as well as GAPDH, specific products were amplified and detected using the cycle profile of the Qiagen SYBR green reagent (Qiagen, Valencia, CA, USA). The sequences of primers used in this study were shown in the Appendix A.

### 2.6. CellMiner Data Mining and Analysis

The CellMiner tool (http://discover.nci.nih.gov/cellminer, accessed on 30 March 2021) was used to compare and plot the relative baseline expression of DPEP1, KRT80, FABP6, NKD2, FOXQ1, CEMIP, ETV4, TESC, FUT1, and GAS2 mRNAs in the National Cancer Institute (NCI)-60 cell line panel. The tool enables the retrieval and integrated analyses of baseline and experimental data compiled from the 60 cell lines included in the panel. We selected gene transcript level z scores for the analysis of the 10 genes as gene identifier inputs.

### 2.7. UALCAN Analysis

The UALCAN website (https://ualcan.path.uab.edu/, accessed on 30 March 2021) is useful for the effective analysis of cancer data in the Cancer Genome Atlas (TCGA) database [13]. The website can be used to analyze genes correlated with the different cancer types, cancer stages, and prognostic factors using the TCGA database samples. We further analyzed the relationship between the expression levels of the top 10 upregulated CRC-associated genes and individual cancer stages using the UALCAN database.

### 2.8. Oncomine Database Analysis

Gene expression changes were analyzed using the TCGA microarray dataset of the Oncomine website (www.oncomine.org, accessed on 30 March 2021) by using colorectal tumors and normal colorectal tissues. In this study, the mRNA expression levels of the top 10 upregulated CRC-associated genes were compared in CRC and normal control samples using the Oncomine database.

### 2.9. Survival Analysis

Survival analysis was performed using the online survival analysis and biomarker validation tool SurvExpress [14]. We evaluated the data of 466 CRC patients from TCGA data (the 28 January 2016 version) to deposit them in the SurvExpress database. SurvExpress classified patient samples into two groups, high risk and low risk, based on the average expression of the 10 genes’ signature values, and each group was determined according to the ordered prognostic index, which is based on the expression levels and values obtained from the Cox fitting algorithm [15].

### 2.10. Gene Ontology, Functional, and Signaling Pathway Analyses

The WEB-based Gene SeT AnaLysis Toolkit (WebGestalt; http://www.webgestalt.org/option.php, accessed on 30 March 2021) [16] was used for the Gene Ontology (GO) analysis of the top 10 upregulated CRC-associated genes. The GO enrichment analyses consisted of a biological process analysis, cellular component analysis, and molecule function analysis. The search tool for the Retrieval of Interacting Genes/Proteins (STRING) database (http://string-db.org/, accessed on 30 March 2021) was used for protein–protein interaction (PPI) predictions [17]. Functional and signaling pathways of the top 10 upregulated CRC-associated genes were analyzed using Enrichr, a public database [18].

### 2.11. cBioPortal Analyses

cBioPortal, an open-access online tool for the multidimensional analysis of cancer genomics, enables the visualization and analysis of genes, including DNA copy number analysis, mRNA, nonsynonymous mutation analysis, and DNA methylation analysis [19]. The colorectal adenocarcinoma (TCGA) dataset, which includes data on 524 patients, was selected for further analysis of the top 10 upregulated CRC-associated genes. The genomic profiles included mutations, putative copy number alterations, and mRNA expression Z scores (RNA Seq V2 RSEM), with a z score threshold of ±2.0.

### 2.12. Gene Expression Profiling Interactive Analysis

The public Gene Expression Profiling Interactive Analysis (GEPIA) database can be used to analyze gene expression profiles [20]. Furthermore, it can be used for the multidimensional analysis of cancers. In this study, the coexpression of CTNNB1 and the top 10 upregulated CRC-associated genes were analyzed in 275 CRC patients with data in the TCGA using Pearson’s correlation analysis, with *p* < 0.05 considered statistically significant.

### 2.13. Analysis of CTNNB1 and Its Corresponding Target Genes in Multiple CRC Clinical Datasets

In silico analysis was performed using the CANCERTOOL [21]. Genes were searched in the publicly available CRC transcriptome datasets that are incorporated into the CANCERTOOL database. Datasets used: Colonomics (https://www.colonomics.org/, accessed on 30 March 2021); Jorissen et al. [22]; Laibe et al. [23], and TCGA (RNA-seq) (https://cancergenome.nih.gov/, accessed on 30 March 2021).

### 2.14. Statistical Analysis

*p* values and fold-changes for the differential expression analysis of genes generated from the CRC tissues in the Oncomine database, UALCAN database, and survival analyses were calculated using a one-sided Student’s *t*-test. *p* values of <0.05 were considered significant.

## 3. Results

### 3.1. mRNA Expression of Top 10 Upregulated CRC-Associated Genes in Various Human Cancer Cell Lines and Tissues

Our recent study [10] identified the top 10 significantly upregulated genes in CRC tissues compared with normal colorectal tissues through the analysis of the GEO dataset (GSE21815) (unpublished data from reference [10]; Appendix A). We found that the levels of the top 10 upregulated genes simultaneously increased by 108.65 to 30.49 times in CRC tissues compared with those in normal colorectal tissues. To further understand the expression levels of these genes in human cancers, we used the CellMiner NCI-60 analysis tool [24] to determine the mRNA expression of these genes in NCI-60 cancer cell lines. Among the 60 cell lines, the top 10 gene transcript levels were highly expressed in approximately half of the CRC cell lines (Figure 1A). Furthermore, to investigate the expression levels of the top 10 upregulated CRC-associated genes in CRC tissues, we analyzed the expression levels of 10 genes by using the UALCAN database [13]. Among 24 cancer types, we found that the mRNA levels of 10 genes were significantly upregulated in colon adenocarcinoma (Figure 1B); this finding is generally consistent with our previous findings (Appendix A). With these results taken together, we concluded that 10 genes were generally overexpressed in various cancers, particularly in CRC.

### 3.2. mRNA Expression and Clinical Relevance of the Top 10 Upregulated CRC-Associated Genes in CRC Tissues and Patients

To confirm the mRNA expression levels of the aforementioned 10 genes in numerous CRC tissues, we analyzed the mRNA expression profiles of 10 genes by using the TCGA dataset of the Oncomine database. Compared with normal samples, the mRNA expression of the 10 genes significantly increased in colorectal tumor tissues. Significant increases were found in the mRNA expression of *DPEP1* (13.47-fold; Figure 2A), *KRT80* (20.02-fold; Figure 2B), *FABP6* (13.69-fold; Figure 2C), *NKD2* (6.80-fold; Figure 2D), *FOXQ1* (46.12-fold; Figure 2E), *CEMIP* (30.80-fold; Figure 2F), *ETV4* (10.55-fold; Figure 2G), *TESC* (7.99-fold; Figure 2H), *FUT1* (5.02-fold; Figure 2I), and *GAS2* (4.72-fold; Figure 2J) in CRC tissues (*n* = 101), compared with normal colon tissues (*n* = 19). In addition, to verify the specificity of 10 genes in CRC tissues, we increased the number of samples and integrated 716 samples from six studies including, Hong, Kaiser, Ki, Sabates-Bellver, Skrzypczak, and TCGA databases in ONCOMINE by meta-analyses. As shown in the Appendix A, high expression of the top 10 upregulated CRC-associated genes was observed in 716 samples from six CRC databases. Subsequently, the relationship between the mRNA expression levels of the 10 genes and the tumor stage related to the 274 TCGA-colon adenocarcinoma samples was analyzed using the UALCAN database. The mRNA expressions of the 10 genes changed significantly across all the tumor stages of CRC (Figure 3). In addition, the expression of the KRT80 (Figure 3B), FABP6 (Figure 3C), NKD2 (Figure 3D), FOXQ1 (Figure 3E), ETV4 (Figure 3G), and GAS2 (Figure 3J) transcripts was significantly higher than in the later stages (stages III and IV) compared to the earlier stages (stages I and II). These findings revealed that the highly expressed of these 10 genes in tumor tissues of CRC patients, and the expression of six genes among the top 10 genes, was higher in the CRC patients with advanced stages.

### 3.3. Prognosis of the Top 10 Upregulated CRC-Associated Genes in CRC Samples Was Analyzed Using Kaplan–Meier Analyses

The prognostic relevance of the 10 genes in CRC patients from the TCGA-CRC dataset (*n* = 466) was analyzed using a SurvExpress survival analysis [25]. The results demonstrated that the high expression levels of *DPEP1* (Figure 4A), *NKD2* (Figure 4B), *CEMIP* (Figure 4C), *ETV4* (Figure 4D), *TESC* (Figure 4E), and *FUT1* (Figure 4F) were associated with poor outcomes in CRC patients. However, the expression levels of KRT80, FABP6, FOXQ1, and GAS2 mRNAs (Appendix A) were not significantly associated with the clinical outcomes of CRC patients. Collectively, these results indicate that a high expression of DPEP1, NKD2, CEMIP, ETV4, TESC, or FUT1 may indicate a worse prognosis for CRC patients.

### 3.4. Molecular Network and Functional Enrichment Analyses of the Top 10 Upregulated CRC-Associated Genes

The 10 upregulated CRC-associated genes were used as input molecules in the WEB-based Gene SeT AnaLysis Toolkit (WebGestalt) [16] to perform GO enrichment analyses (Figure 5A). In the biological process category, the top three categories for the 10 upregulated CRC-associated genes were as follows: (i) metabolic process (eight genes), (ii) biological regulation (eight genes), and (iii) localization (five genes). In terms of cellular components, of the 10 genes, seven were located in the membrane and five were located in the nucleus. In the molecular function category, seven of the 10 genes were associated with protein binding. We performed PPI and pathway enrichment analyses. For the PPI analysis, we used the 10 upregulated CRC-associated genes in the search tool for the Retrieval of Interacting Genes/Proteins (STRING) database [17]. No mutual interaction was observed between these 10 molecules (Figure 5B). Furthermore, pathway enrichment analysis was performed using the Enrichr method [18]. NCI-Nature enrichment indicated that the canonical Wnt signaling pathway was the main one involved in the 10 upregulated CRC-associated genes’ network signaling (Figure 5C). To further investigate the molecules involved in the canonical Wnt signaling pathway in the interaction among the 10 upregulated CRC-associated genes, a PPI network was constructed from the STRING database. Surprisingly, the PPI network demonstrated that the key gene involved in the canonical Wnt signaling pathway for interaction with the 10 upregulated CRC-associated genes was CTNNB1 (Figure 5D).

### 3.5. Top 10 Upregulated CRC-Associated Genes in Genetic Alterations and Their Correlation with CTNNB1 in CRC

Studies have demonstrated that mutations in the *APC* gene are detected in approximately 70%–80% of human colon tumors, indicating that the activation of the Wnt/beta-catenin (*CTNNB1*) pathway plays a crucial role in human colon tumorigenesis [25]. Thus, we first explored the genetic alterations of *APC*, *CTNNB1*, and the top 10 upregulated CRC-associated genes through the cBioPortal database [19], which includes the data of 524 CRC patients from the TCGA. The percentages of genetic alterations of *APC*, *CTNNB1*, and the top 10 upregulated CRC-associated genes in CRC were 69% for *APC*, 13% for *CTNNB1*, 8% for *DPEP1*, 7% for *KRT80*, 6% for *FABP6*, 7% for *NKD2*, 6% for *FOXQ1*, 8% for *CEMIP*, 8% for *ETV4*, 7% for *TESC*, 6% for *FUT1*, and 8% for *GAS2*, respectively (Figure 6A). Subsequently, the GEPIA database, comprising data of 275 CRC patients from the TCGA [20], was used to investigate the correlation between the expression of CTNNB1 and the top 10 upregulated CRC-associated genes. CTNNB1 was positively correlated with DPEP1 (R = 0.34, *p* < 0.001; Figure 6B), KRT80 (R = 0.16, *p* = 0.01; Figure 6C), NKD2 (R = 0.21, *p* < 0.001; Figure 6D), FOXQ1 (R = 0.24, *p* < 0.001; Figure 6E), CEMIP (R = 0.35, *p* < 0.001; Figure 6F), FUT1 (R = 0.13, *p* < 0.05; Figure 6G), and GAS2 (R = 0.32, *p* < 0.001; Figure 6H). However, there was no correlation between CTNNB1 and FABP6, ETV4, or TESC in 275 CRC patients from the TCGA database (Appendix A). Taken together, these data provide evidence that DPEP1, KRT80, NKD2, FOXQ1, CEMIP, FUT1, GAS2, and CTNNB1 have a regulation correlation in CRC.

### 3.6. CTNNB1 Plays the Central Role to Regulate NKD2, FOXQ1, and CEMIP Expression

To further investigate the correlation between CTNNB1 and the aforementioned seven genes, we first investigated the expression of CTNNB1 and the aforementioned seven genes in SW480 cells with CTNNB1 knockdown compared with the siRNA control-transfected SW480 cells, which were obtained from the public GEO dataset (GSE20966). We observed that the CTNNB1 transcript was downregulated by approximately 1.5 times in siCTNNB1-transfected SW480 cells and the expression trend of NKD2, FOXQ1, and CEMIP transcripts was also downregulated (Figure 7A). Next, we performed the web-based interface-CANCERTOOL [21] for the correlation analyses of CTNNB1 and corresponding target genes of CTNNB1 in multiple CRC clinical datasets. As shown in Figure 7B, a positive correlation between the expression of CTNNB1 and NKD2, FOXQ1, or CEMIP was observed in the four available CRC datasets. To address the abovementioned, the expression of the CTNNB1, NKD2, and FOXQ1 transcripts was detected in siCTNNB1-transfected HCT116 cells. As shown in Figure 7C–E, the expression of NKD2, FOXQ1, and CEMIP transcripts were significantly downregulated in the CTNNB1-knockdown of HCT116 cell line. In addition, in order to understand the functions of NKD2, FOXQ1, and CEMIP genes in CRC cells, the viability of CRC cells was manipulated through NKD2, FOXQ1, and CEMIP expression, respectively. Compared to the control siRNA-transfected HCT116 cells, knockdown of the endogenous expression of NKD2, FOXQ1, or CEMIP in HCT116 cells caused significant decreases in cell proliferation (Figure 7F for siNKD2, Figure 7G for siFOXQ1, Figure 7H for siCEMIP) and colony numbers and sizes, as compared to the control siRNA (Figure 7I for siNKD2, Figure 7J for siFOXQ1, Figure 7K for siCEMIP). Collectively, our results suggest that CTNNB1 plays a critical role in regulating the expression of NKD2, FOXQ1, and CEMIP in CRC cells.

## 4. Discussion

An increase in both the number of biomarkers and their uses helps with prognostication and treatment decision-making for CRC [26]. In addition, with recent advances in medicine and biotechnology, precise treatment of CRC through gene-targeted therapy is a new and efficient therapeutic approach for CRC, which requires the identification of marker genes that are associated with cancer [27]. This study identifies that CTNNB1 is the key factor in the interaction of the canonical Wnt signaling pathway with 10 upregulated CRC-associated genes. Several lines of evidence support this finding. Firstly, we demonstrated that the top 10 upregulated CRC-associated gene (*DPEP1*, *KRT80*, *FABP6*, *NKD2*, *FOXQ1*, *CEMIP*, *ETV4*, *TESC*, *FUT1*, and *GAS2*) transcripts were upregulated in CRC cell lines and public CRC datasets. Secondly, we found that the top 10 upregulated CRC-associated genes affected the prognosis of CRC patients, and six of the genes had high expression in the CRC subgroup with more severe malignancy. Thirdly, we found that a high expression of DPEP1, NKD2, CEMIP, ETV4, TESC, or FUT1 was significantly correlated with worse prognosis in CRC patients. Fourthly, the PPI network demonstrated that CTNNB1 is the key factor involved in the interaction of the canonical Wnt signaling pathway with the 10 upregulated CRC-associated genes. Fifthly, we demonstrated *NKD2*, *FOXQ1*, and *CEMIP* as three CTNNB1-regulated genes, and individual inhibition of the expression of the three CTNNB1-regulated genes can cause growth inhibition of CRC cells. Collectively, this study identifies three putative CTNNB1-regulated genes from the top 10 upregulated CRC-associated genes and provides a new molecular interaction network for CRC.

The aberrant Wnt/beta-catenin (CTNNB1) signaling pathway plays a crucial role in CRC development [25]. In the canonical Wnt cascade, beta-catenin is the key effector responsible for the transduction of the signal to the nucleus, and it triggers the transcription of Wnt-specific genes responsible for the control of cell proliferation [28]. Mutations in APC occur in approximately 70%–80% of human colon tumors, which activate the Wnt/beta-catenin pathway to induce beta-catenin to form complexes with the TCF/LEF nucleus partner in the nucleus, subsequently regulating the expression of beta-catenin target genes [29]. In our study, we demonstrate that *NKD2*, *FOXQ1*, and *CEMIP* are three CTNNB1-regulated genes. In addition, through cellular component analysis, we found that these three genes were located in the nucleus. Furthermore, NKD2 [30] and FOXQ1 [31] proteins have nuclear localization signals, which can tag the protein for import into the nucleus through nuclear transport. For CEMIP protein, although it lacks a nuclear import signal, strong CEMIP expression was detected in the nucleus of CRC tissues. Moreover, nuclear CEMIP protein expression is positively correlated with the expression of nuclear beta-catenin in colon adenocarcinomas tissues [32]. Based on these findings, CTNNB1 may regulate the expression of these three genes by transcriptional regulation.

Biomarkers for cancers are mostly determined through the identification of significant differentially expressed genes in high-throughput studies of cancer [33]. Biomarkers, such as miRNAs [34], long non-coding RNAs (lncRNAs) [35], and DNA methylation hotspots [36] have been demonstrated to associate with CRC tumor development. The TCGA and GEO databases have a large number of mRNA expression data of tumor samples related to multiple cancers [37]. However, most related studies have not had a sufficient sample size, and some of these studies included samples of only one database, which may lead to imprecise results [38]. Furthermore, few studies have used different databases to verify their own data of patients to identify biomarkers of CRC, which may be a reliable and precise approach to cancer research. Our recent study identified the top 10 upregulated genes in 132 CRC tissues compared with nine normal colorectal tissues by using the GEO database (GSE21815) (unpublished data from [10]). In the current study, we further investigate the expression levels of the top 10 upregulated CRC-associated genes in 101 CRC tissues compared with 19 normal colon tissues by using the Oncomine database. In addition, the mRNA expression levels of the 10 genes are analyzed in numerous CRC tissues by using TCGA data through UALCAN, which contains 274 CRC tissues. In addition, the prognostic relevance of the top 10 upregulated CRC-associated genes is analyzed in 466 CRC patients whose clinical data are available in the TCGA-CRC data set. Furthermore, we used the GEO database (GSE14333) to perform univariate and multivariate COX regression analyses to assess the top 10 upregulated CRC-associated gene in predicting survival outcome of CRC patients. We access the Prognoscan database [39], collect the data from GSE14333, and contain 226 colorectal cancer (CRC) samples. The patients are divided into two (high and low) expression groups at the all potential cut-point, and the risk differences of the two groups are estimated by a log-rank test. As shown in the Appendix A, among the top 10 upregulated CRC-associated gene, only three genes inversely correlated with survival are identified (*p* < 0.01). Subsequently, the multivariate stepwise Cox regression analysis of the top 10 upregulated CRC-associated genes was applied and three genes are eventually proved to be correlated with overall survival (Appendix A). Moreover, genetic alterations of the top 10 upregulated CRC-associated genes are analyzed in 524 (TCGA-CRC) patients by using the cBioPortal database. Finally, the expression correlation between CTNNB1 and the top 10 upregulated CRC-associated genes is investigated using the GEPIA database with 275 TCGA-CRC patients. According to the aforementioned functional database analysis, we use different databases to verify the expression correlation between the top 10 upregulated CRC-associated genes and individual cancer stages of CRC, prognostic relevance, and molecular network of CTNNB1, which may be a reliable and precise approach to cancer research.

In this study, we identify the potential biomarkers that offer new avenues for diagnosis and prognosis of CRC. In particular, we identified *NKD2*, *FOXQ1*, and *CEMIP* as CTNB1-regulated genes, and the expression levels of these genes may be used as surrogate markers for indicating the activated Wnt/beta-catenin pathway in CRC tissues.

## 5. Conclusions

In this study, the mRNA expression levels, prognostic value, genetic alteration, and expression correlation of the top 10 upregulated CRC-associated genes in CRC are systemically analyzed. The results show that the expression levels of the top 10 upregulated CRC-associated gene transcripts (DPEP1, KRT80, FABP6, NKD2, FOXQ1, CEMIP, ETV4, TESC, FUT1, and GAS2) relate with CRC prognosis, and six of the genes (KRT80, FABP6, NKD2, FOXQ1, ETV4, and GAS2) are highly expressed in the CRC patients in advanced stages. CTNNB1 is the key factor involved in the interaction of the canonical Wnt signaling pathway with 10 upregulated CRC-associated genes. We identified NKD2, FOXQ1, and CEMIP as three CTNNB1-regulated genes and and individual inhibition of the expression of the three CTNNB1-regulated genes can cause growth inhibition of CRC cells. However, further experiments should be conducted to verify the regulatory mechanism between CTNNB1 and the three aforementioned CTNNB1-regulated genes.

## Figures and Tables

**Figure 1 biomedicines-09-01331-f001:**
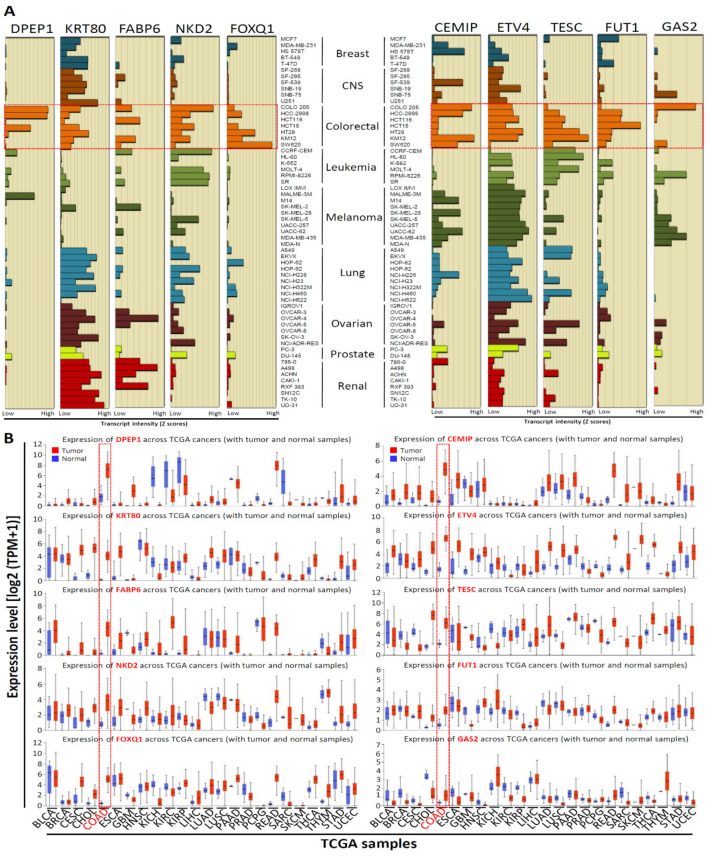
Top 10 upregulated CRC-associated genes mRNA expression in the National Cancer Institute (NCI)-60 human tumor cell lines and TCGA pan-cancer tissues. (**A**) Relative top 10 upregulated CRC-associated genes expression profile in National Cancer Institute (NCI)-60 cancer cell lines from the CellMiner NCI-60 analysis tool. Bars to the left show relative low expression, while bars to right show high expression. (**B**) The top 10 upregulated CRC-associated genes expression in various clinical cancer tissues were compared with those in corresponding normal tissues (UALCAN database), the red dot rectangle indicated colon adenocarcinoma (COAD).

**Figure 2 biomedicines-09-01331-f002:**
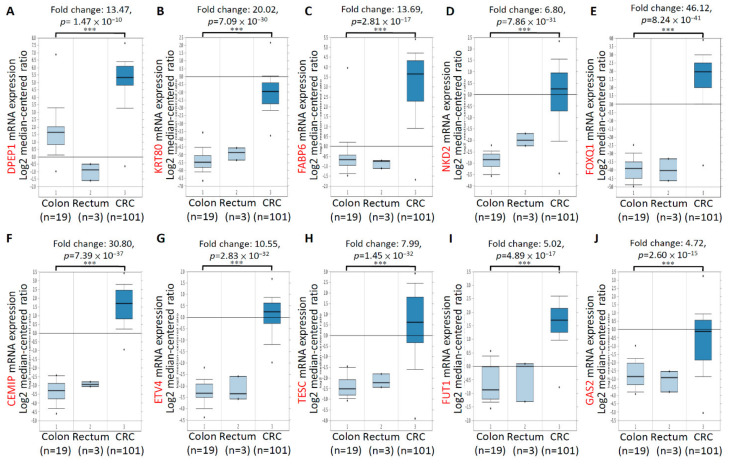
Upregulation of mRNA expression of top 10 upregulated CRC-associated genes using the Oncomine database analysis. The significant increases in the mRNA expression of DPEP1 (**A**), KRT80 (**B**), FABP6 (**C**), NKD2 (**D**), FOXQ1 (**E**), CEMIP (**F**), ETV4 (**G**), TESC (**H**), FUT1 (**I**), and GAS2 (**J**) in CRC tissues (*n* = 101) compared to normal colon tissues (*n* = 19). *** *p* < 0.001.

**Figure 3 biomedicines-09-01331-f003:**
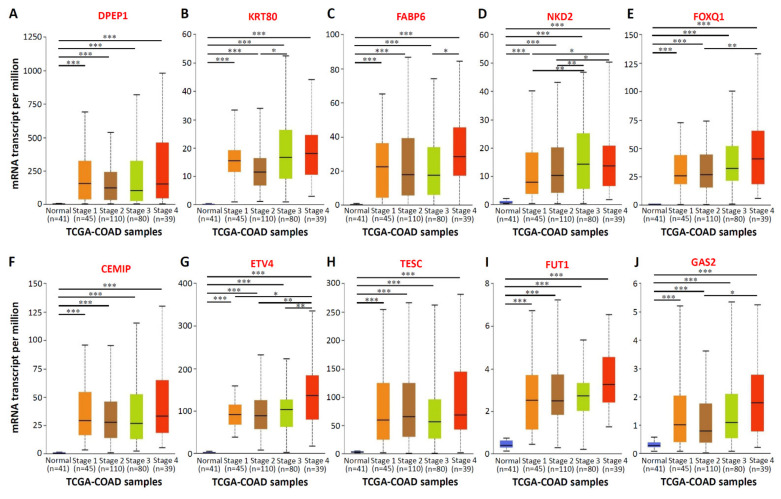
The relationship between mRNA expressions of the top 10 upregulated CRC-associated genes and the individual cancer stages. Individual cancer stages from 274 TCGA-colon adenocarcinoma (COAD) samples include stage 1 to stage 4. The figure presented in (**A**–**J**) to show the mRNA expressions of 10 upregulated CRC-associated genes changed across all the tumor stages of COAD samples. * *p* < 0.05, ** *p* < 0.01, *** *p* < 0.001.

**Figure 4 biomedicines-09-01331-f004:**
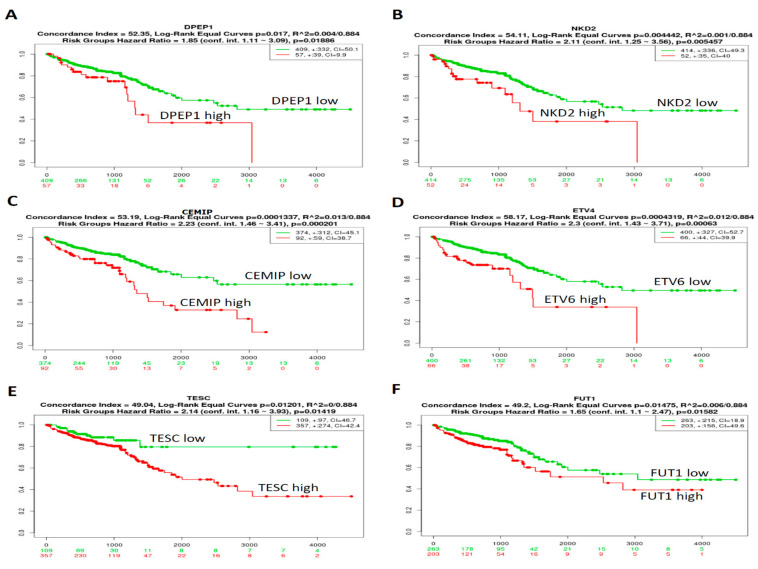
The prognostic value of mRNA expression level of CRC-associated genes in patients with CRC. The relationship between DPEP1 (**A**), NKD2 (**B**), CEMIP (**C**), ETV4 (**D**), TESC (**E**), and FUT1 (**F**) expression and survival status in 466 TCGA-CRC patients. *p* < 0.05 was considered to be statistically significant.

**Figure 5 biomedicines-09-01331-f005:**
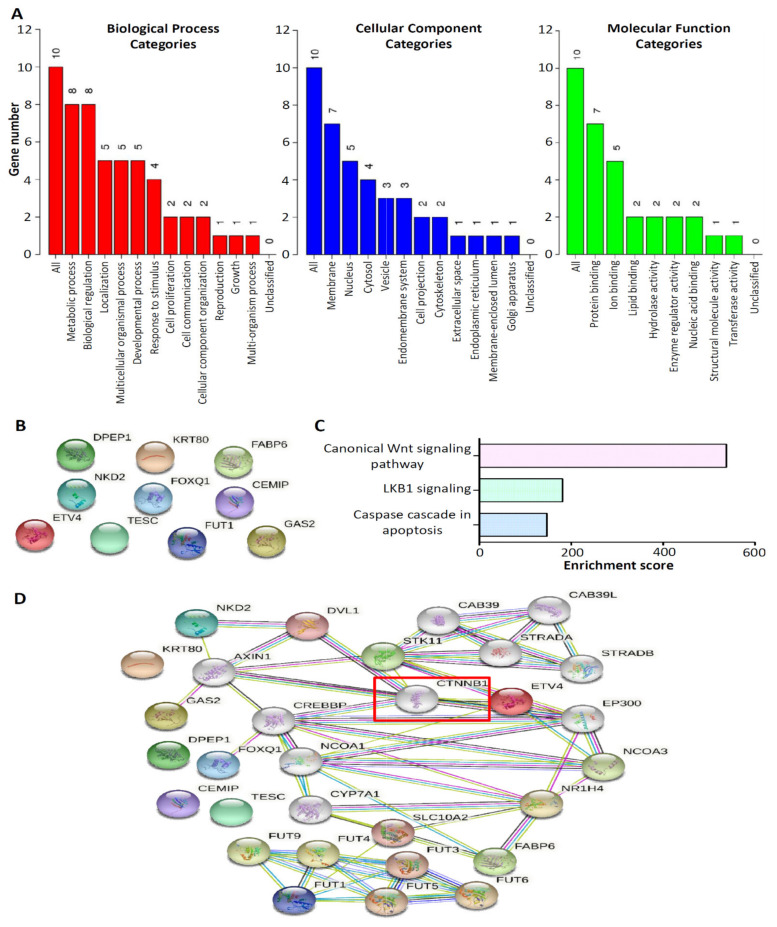
Gene function and pathway enrichment analysis of the top 10 upregulated CRC-associated genes. (**A**) Gene ontology analysis of the top 10 upregulated CRC-associated genes. The results are presented for the following three categories: Biological process, cellular component, and molecular function. The bar represents the number of genes in the user list. Protein–protein interaction network analysis of the top 10 upregulated CRC-associated genes (**B**), which was involved in the canonical Wnt signaling pathway to perform using the Enrichr method (**C**). (**D**) CTNNB1 was predicted as the key factor involved in the canonical Wnt signaling pathway for interaction with the 10 upregulated CRC-associated genes.

**Figure 6 biomedicines-09-01331-f006:**
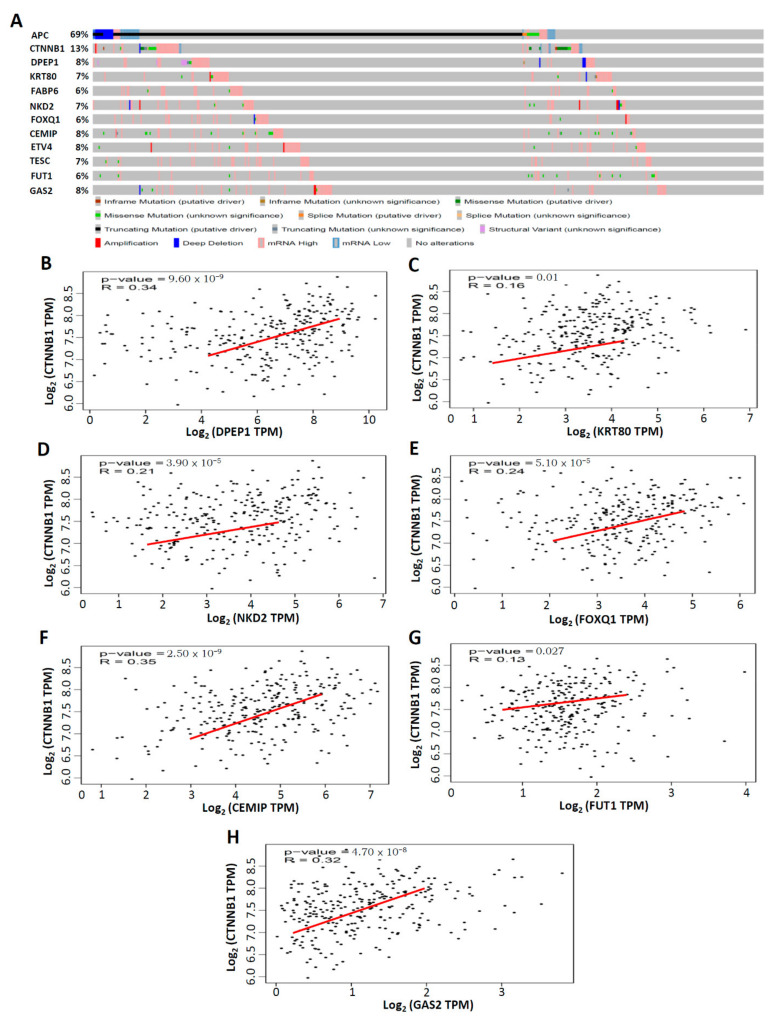
Mutations of *APC*, *CTNNB1*, and the top 10 upregulated CRC-associated genes and the correlation between CTNNB1 and seven CRC-associated genes in CRC. (**A**) *APC*, *CTNNB1*, and top 10 upregulated CRC-associated genes mutation analysis and gene expression in CRC (cBioPortal). (**B**–**H**) The positive correlation between CTNNB1 and seven CRC-associated genes in CRC, analyzed by GEPIA database. *p* < 0.05 was considered to be statistically significant.

**Figure 7 biomedicines-09-01331-f007:**
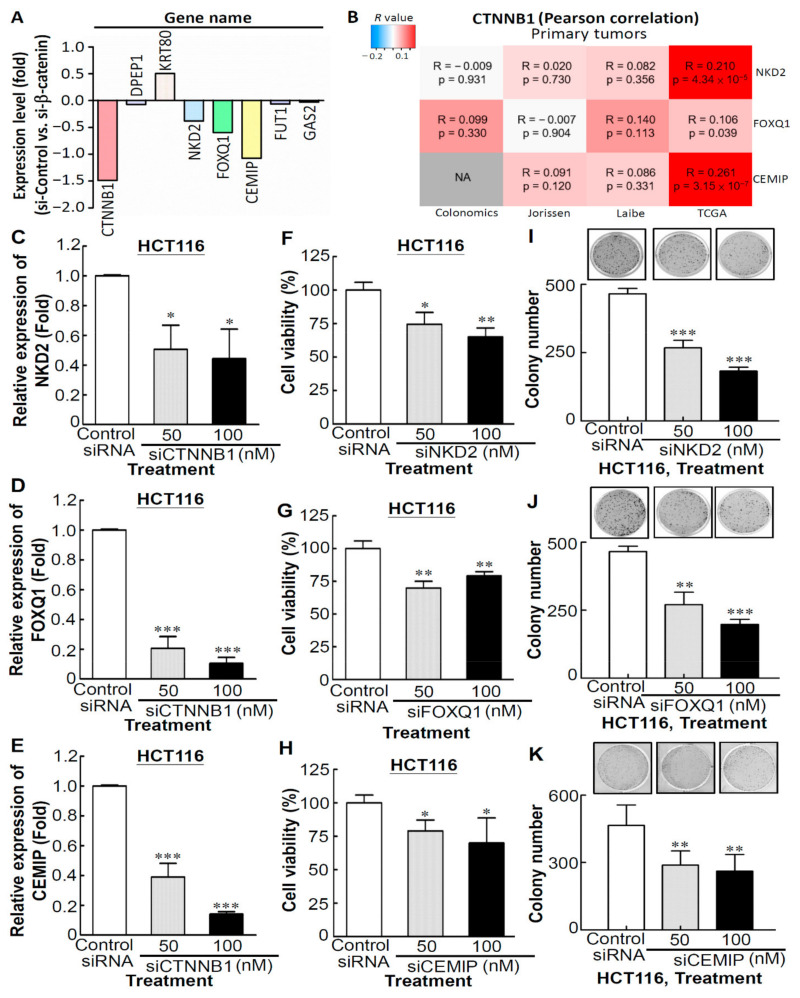
CTNNB1 plays a critical role in regulating the expression of NKD2, FOXQ1, and CEMIP in CRC cells (**A**) CTNNB1 and seven positive expression correlation genes transcripts in CTNNB1 knockdown SW480 cells. (**B**) Heatmap showing the correlation between CTNNB1 and corresponding target genes of CTNNB1 in multiple CRC clinical datasets. Red color indicates positive correlation, and blue color indicates negative correlation, with color intensity representing the Pearson’s correlation value (R), as indicated by the scale on the upper left corner. (**C**–**E**) Expressions of representative genes (*NKD2*, FOXQ1, and CEMIP) in CTNNB1-knockdown HCT116 cells. Cell viability assay was performed to determine the proliferation of NKD2 (**F**), FOXQ1 (**G**) or CEMIP (**H**)-depleted HCT116 cells. The colony-forming assay was performed to determine the effects of NKD2 (**I**), FOXQ1 (**J**), or CEMIP (**K**)-depletion of the growth of HCT116 cells. ** p <* 0.05, *** p <* 0.01, **** p <* 0.001.

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
