# Peer review of "Expression Profile and Prognostic Value of Wnt Signaling Pathway Molecules in Colorectal Cancer"

_biomedicines, 2021, doi:10.3390/biomedicines9101331_

Round 1

Reviewer 1 Report

Dear authors,

the experimental workflow was well designed, the study was well written and the conclusions are easy to understand and supported by the results. According to me, this work could be relevant for the scientific community. To improve the quality of your work, some points should be addressed before publication.

Main points:
A) To show clearly the results, only statistically significant results should be reported in the main figure. Figure 4 should be splitted in two: Panel A, D, F, G, H, I in the main figure and panels B, C, E and J will be moved in a new supplementary figure.
B) The panels B-K of the figure 6 should be splitted in two figures for the same reason of the point (A. In this case, the panels were enlarged to fit in a page, please, adjust, resize the panels correctly and use the same rule of point (A: only statistically significant data should be shown in the main figure and not-statistically significant data should be reported in a new supplementary figure.
C) As reported in the discussion paragraph, "...further experiments should be conducted to verify the regulatory mechanism between CTNNB1 and the CTNNB1-regulated genes." , this warning should be reported also in the "Conclusions" paragraph.

Minor points
1) Maybe the supplementary file was a "zip" file renamed as "rar" file. Check the supplementary file please, I had some problems to correctly unzip it.
2) Gene Symbols in Table S2 should be written in Italic form.
3) Add ref [10] at the beginning of the sentence when you reported "Our recent study [REF] showed" at both lines 11 and 37.

Have a nice day and best regards

Author Response

We appreciate the comments of the Reviewer and believe that our manuscript has been improved by attention to him or her. Our responses to the specific issues raised by the Reviewer are shown in the attachment.

Reviewer 2 Report

none to be made

Author Response

Manuscript ID: biomedicines-1387369
Type of manuscript: Article
Title: Expression Profile and Prognostic Value of Wnt Signaling Pathway Molecules in Colorectal Cancer

Reviewer Comments

Reviewer 2

 We appreciate the comments of the Reviewer and believe that our manuscript has been improved by attention to him or her. The followings are our responses to the specific issues raised by the Reviewer:

Point 1: None to be made

Response 1: Thanks for reviewer's kind comment.

Reviewer 3 Report

General

The authors present a large and well performed molecular study of a network of upregulated m RNAs in normal colonic tissues and colorectal carcinoma.

The data presented are of high scientific value and of high interest.

The data are well documented and precisely explained.

However, all conclusions drawn by the authors – from a clinical view point- are misleading.

The authors show that the expression of the investigated biomarkers are correlated with the stages of the tumors. However, how come the authors to the conclusion, that this phenomenon my serve as a base for “an accurate prediction of the optimal therapeutical intervention for CRC patients”? From a clinical view an improvement of the “therapeutical intervention” would be a prognostic information beyond the information which stage can give (especially at stage 3) or, additionally, a new target for systemic treatment. Both aspects are not presented by the authors.

This point also refers to the correlation with the “diagnosis” of CRC (line 47) and “personalized therapy”.

The authors should find a much more conclusive interpretation of their results from a clinical view point and omit all misleading conclusions about screening, diagnosis, prognostication, selection of optimal therapeutical interventions and personalized treatment.

Specific

Line 25: increasing instead “in-creasing”

Line 29: 10 “upregulated genes are upregulated” ?

Line 25: inhibition of the

Line 36: omit “the”

Line 37: for

Line 46: Detection of CRC still is mainly based on FOBT and coloscopy

Author Response

(The authors gave the same response as above.)

Round 2

Reviewer 3 Report

the authors follow all recommendations very carefully